# Post-Mortem Dental Profile as a Powerful Tool in Animal Forensic Investigations—A Review

**DOI:** 10.3390/ani12162038

**Published:** 2022-08-10

**Authors:** Joan Viciano, Sandra López-Lázaro, Carmen Tanga

**Affiliations:** 1Department of Medicine and Ageing Sciences, ‘G. d’Annunzio’ University of Chieti-Pescara, 66100 Chieti, Italy; 2Departamento de Antropología, Facultad de Ciencias Sociales, Universidad de Chile, Santiago 6850331, Chile; 3Forensic Dentistry Lab, Centro de Investigación en Odontología Legal y Forense–CIO–, Facultad de Odontología, Universidad de La Frontera, Temuco 4811230, Chile; 4Department of Legal Medicine, Toxicology and Physical Anthropology, University of Granada, 18016 Granada, Spain

**Keywords:** veterinary forensics, carcasses, teeth, species identification, sex estimation, age-at-death estimation, body size estimation, geographical origin, post-mortem interval estimation, bite mark analysis

## Abstract

**Simple Summary:**

Teeth are the hardest anatomical structure of the animal body. As a result, even when preservation conditions are extremely poor and the rest of the skeleton decomposes, the dentition is often still preserved. The strong nature of the teeth means that they are often an invaluable biological source of information about a deceased animal. This is particularly important in forensic investigations resolving legal disputes involving animals and in circumstances where the animal body is recovered a long time after death. The post-mortem dental profile can contribute information such as species identification, sex, age-at-death, body size, geographical origin (provenance), and post-mortem interval. Although the dental profile may not lead to a positive identification, it can narrow the pool toward a presumptive identity. In this review, we briefly examine different dental techniques using characteristics of teeth as a means of identification of freshly deceased and skeletonised animals, highlighting the importance of dentition in the identification process in forensic contexts.

**Abstract:**

Veterinary forensics is becoming more important in our society as a result of the growing demand for investigations related to crimes against animals or investigations of criminal deaths caused by animals. A veterinarian may participate as an expert witness or may be required to give forensic assistance, by providing knowledge of the specialty to establish a complete picture of the involvement of an animal and allowing the Courts to reach a verdict. By applying diverse dental profiling techniques, not only can species, sex, age-at-death, and body size of an animal be estimated, but also data about their geographical origin (provenance) and the post-mortem interval. This review concentrates on the dental techniques that use the characteristics of teeth as a means of identification of freshly deceased and skeletonised animals. Furthermore, this highlights the information that can be extracted about the animal from the post-mortem dental profile.

## 1. Animals in Forensic Sciences

In a broader sense, veterinary forensics can be defined as the application of veterinary science in resolving legal disputes involving animals (i.e., livestock, wild, exotic, and household animals) and animal derivatives [1]. Animals may be involved in two diverse ways: they may either be the victim of an assault or illegal act (i.e., maliciously or accidentally ‘human-induced’ injuries and/or insults to animals), or the perpetrator when the animal causes the incident (i.e., injuries caused to humans) [2]. This discipline is becoming more important in our society, increasing its frequency worldwide as a result of the growing demand for investigations related to crimes against animals or investigations of criminal deaths caused by animals [3]. The main fields in which a veterinarian may participate as an expert witness or may be required to provide forensic assistance are the following [4,5,6]:*Assessment of animal welfare* (includes survival-related factors: nutrition, environment, health; situation-related factor: behaviour; emotional-related factor: mental state). Includes giving an opinion as to whether an animal may be experiencing or has experienced pain, discomfort, or distress in the past. In some instances, this evaluation may incorporate experts from other scientific disciplines, such as nutritionists and animal behaviourists.*Determination of the time, the cause, and the circumstances of death of an animal, as well as other related investigations*, such as the identification and interpretation of changes in the different body tissues, recognition of parasites and detection of signs of poisoning.*Verification of the history and provenance of live/dead animals and animal derivatives*. Most of the cases are related to domestic species; however, other cases fall under the category of ‘wildlife crime’. In this latter case, this field is related to whether national or international conservation legislation has been breached, such as the Convention on International Trade in Endangered Species of Wild Fauna and Flora (CITES). This is a large field that includes offences and trapping of wild animals for sale, malicious poisoning, poaching, and the importation of endangered species or derivatives thereof.*Performing clinical or post-mortem examinations when animal abuse appears to be related to acts of violence towards humans.**Food safety.* Ranges from animal welfare during slaughter (including religious slaughter) to meat inspection standards and the detection of contaminants, import, export, and correct identification of food.*Human welfare.* Covers public health issues such as zoonotic infections, health and safety (both in the workplace and in care institutions), and personal injury cases (e.g., kicks, bites, venomous bites, scratches, etc.).*Miscellaneous.* Miscellaneous cases relating to legal responsibility, negligence, nuisance, fraud, environmental pollution, damage to Crown property (e.g., the swans of Queen Elizabeth II in the UK), as well as other miscellaneous legislation and situations.

### 1.1. The Animal as the Victim

There are many ways to inflict injuries and/or insults on animals maliciously or accidentally, including forms of physical, sexual, and psychological abuse [2,7]. Physical injuries may be the result of trauma events, excess heat or cold, immersion in water or other insults; these injuries are usually unintentional but may include non-accidental injuries. Injuries resulting from a sexual insult may be the result of attempted animal sexual abuse, or surgical or malicious damage of the urogenital region (including castration). These injuries can be the consequence of true sexual abuse or also due to normal veterinary/husbandry practices. The third insult, not legally accepted in some instances, is psychological in nature, which may be the result of taunting, teasing, or threatening an animal, or the deprivation of companionship or inappropriate social grouping. Recent behavioural studies conducted by forensic animal behaviourists document these forms of psychological effects [8], as well as the cultural conditions determining the manner of killing.

There have been extensive studies of some forms of ‘human-induced’ damage to animals; for example, non-accidental injury caused to dogs and cats [9,10], the effects of traps and snares [11,12], poisoning [13,14], and shooting on wildlife [12,15]. Death, injury, health concerns, pain, or distress may result from most of the examples cited above but the implications may differ depending on the species and the circumstances surrounding the attack [2,16]. Kellert and Felthous [17] proposed nine distinct motivations for animal cruelty when questioning groups of aggressive criminals, non-aggressive criminals, and non-criminals about their past experiences with animals:*To control an animal.* Excessive and cruel physical punishment may be employed to exert control or shape an animal’s undesirable behaviours.*To retaliate against an animal.* Extreme punishment or retaliation may be inflicted for suspected misbehaviour on the part of an animal.*To satisfy a prejudice against a species or breed.* On numerous occasions, people designate certain groups of animals as good or bad. These beliefs may be related to cultural values, such as prejudice in our society against spiders or snakes.*To express aggression through an animal.* Inculcating violent tendencies in the animal in order to express violent, aggressive behaviours toward other people or animals.*To strengthen one’s own aggressiveness.* Killing and abusing animals may be a way to enhance one’s aggressive aptitudes or to impress others with the capacity for violence.*To impress people for amusement.* Cruelty toward animals sometimes occurs as a means of creating amusement and ‘entertaining’ friends.*To retaliate against another person.* Sometimes cruelty toward animals occurs as revenge against other people.*To divert hostility from a person to an animal.* This deviated aggression usually involves authority figures whom the subject hates or fears but who they are afraid to aggress against. For example, in childhood, it is often easier to be violent to an animal than against a parent or adult*Non-specific sadism.* This violence is related to the desire to inflict harm, suffering, or death on an animal in the absence of any particular provocation or especially hostile feelings toward an animal.

### 1.2. The Animal as the Perpetrator

The human–animal interaction dates back to prehistoric times, with animal domestication being practiced for thousands of years [18,19]. From ancient times, humans exploited animals for multiple reasons, generally having productive outcomes for humans, such as farming, obtaining food products (e.g., chickens as suppliers of eggs, pigs and poultry as meat sources), transportation of people or cargoes (e.g., mules for transport of supplies, horses as vehicles), recreation and entertainment (e.g., circus shows, horse racing), sports and hunting (e.g., sporting dogs) [20]. However, interactions between humans and animals can also be unproductive and unintentional, sometimes leading to injuries.

Animals may cause harm to humans or disrupt human activities in many ways, including: physical damage to property (e.g., cattle knocking down fences, birds destroying crops); causing noise (e.g., barking dogs, roosters crowing); producing unpleasant odours (e.g., a piggery or poultry house in the vicinity of a residential area); causing overwhelming fear (e.g., fear of spiders (arachnophobia) or dogs (cynophobia)); causing allergic reactions (e.g., hypersensitivity to animal derivatives, such as fur or feathers); and infecting humans with pathogens that cause a variety of zoonotic diseases (e.g., rabies, avian influenza) [2]. However, the continuous human–animal relationships in urban areas and the increasing human expansion into the animal’s natural habitat, have led to an increased possibility of an animal attack on a human being causing serious injuries and even death [21].

The nature and incidence of animal attacks on humans varies between different regions of the world depending on the fauna present and the extent of the interactions between humans and animals. Many animals have been reported to attack and bite living humans, with most attacks being caused by the order of carnivorous mammals such as canids (e.g., dog, wolf; [22,23]), felids (e.g., cat, lion, tiger; [24,25]), ursids (e.g., brown bear, polar bear; [26,27]), and non-human Primates (e.g., chimpanzee, gorilla, macaque; [28,29]), as well as by ungulated mammals such as suids (e.g., domestic pig, wild boar; [30,31]) and hippopotamids (e.g., hippopotamus; [32]), by rodent mammals (e.g., rat, squirrel; [33,34]), by reptiles (e.g., crocodile, iguana; Komodo dragon; [35,36,37]), and even by sharks (e.g., white shark, tiger shark; [38,39,40]), among other animals.

Similarly, animal scavenging is also relatively frequent in forensic investigations, where animal activity in outdoor settings is one of the main taphonomic agents that significantly affects the preservation of a human corpse when recovered from the area of deposition [41] (see Section 3.4. Bite Marks for more details on this topic). Thus, under varying conditions, any animal can attack a human corpse [42]. Generally, this is a scavenging process whereby the animal produces transportable units from the remains that can be moved to another place for later consumption [43,44], causing post-mortem modifications to a human corpse, altering characteristics of peri-mortem trauma, influencing decomposition rates, disarticulating and scattering body parts, mimicking or destroying actual forensic evidence, and affecting identification of the deceased [41].

## 2. Role of the Veterinary Pathologist in Veterinary Forensic Investigations

The work of the forensic medical pathologist and the forensic veterinary pathologist is similar; however, there is an enormous difference: while the work of the former focuses on a single species (the human being), the work of the latter encompasses multiple species, with cases involving household animals (including exotic species), farm animals, and wild animals. In this way, multispecies forensic pathology makes it a complex and difficult discipline to manage [45]. The forensic veterinary pathologist is not only specifically concerned with the post-mortem examination of a deceased animal and documents the findings of the examination but is also involved in the collection of evidence and court proceedings.

In veterinary forensics, the identification of carcasses is of less importance compared to its counterpart in human forensic medicine, although the reliable identification of live animals can be crucial (e.g., in the resolution of criminal investigations where the animal is the causative agent of the injuries or death of a human being). However, when it is necessary to identify dead animals or their remains, the following methods can be used [2]: (i) external markings, colour patterns, etc.; (ii) external morphological features (e.g., shape of antlers, abnormal coloration, or wear of hooves); (iii) presence of external collars, chains, ear tags, and other human-introduced devices (e.g., transponders); (iv) surgical evidence (e.g., docked tail, prosthesis); and (v) osteological characteristics. In the latter case, the ultimate goal of analysing a set of skeletal remains is to estimate the biological profile (i.e., to establish a set of characteristics that an animal specimen possessed during their life), which can be used to determine identity after death. In veterinary science, the biological profile would include the taxonomic classification (i.e., class, order, family, genus, and species identification), sex, age-at-death, body size, health/disease status, and individualising characteristics [46].

The present review concentrates on the dental methods that use the characteristics of teeth as the means of identification of fresh deceased and skeletonised animals. The review is an attempt to highlight the importance of dentition in the identification process and its utility in estimating the biological profile and to show other information that can be extracted about the animal from the post-mortem dental profile. Figure 1 illustrates the integration of veterinary medicine within the forensic sciences, summarizing the main applications of dental profile in veterinary forensics.

## 3. Teeth as a Biological Source for Forensic Identification in Animal Remains

The distinct anatomy of the dentition and its resistance to decomposition makes it an invaluable source for biological studies and enables us to understand ancient and modern animal communities. Examination of the dentition is widely used by zooarchaeologists to identify animal skeletal remains [47,48], but it is also important in post-mortem forensic work. It is reported that the nineteenth-century French naturalist and zoologist Georges Cuvier, who established the sciences of comparative anatomy and palaeontology, said: ‘*Show me your teeth and I will tell you who you are*’ (translated from French; [49]). Animals’ teeth are so varied and distinctive that they can be used to identify animal remains by veterinary forensics based on a single tooth.

### 3.1. Biological Profile

#### 3.1.1. Species Identification

The comparative dental anatomy analysis is a classical technique for species identification, and it also correlates to the inter-species relationship among members of the same family (e.g., family of Felidae: includes cheetah, leopard, tiger, domestic cat, lynx, among others) [50,51]. The number and types of teeth present in the oral cavity is useful in genus identification. The number of teeth of each type, present in one maxillary or mandibular hemiarch, is referred to as the *dental formula*. For example, the dental formula for the permanent dentition of the family of Canidae (i.e., all dogs, wolves, foxes, and coyotes) is I3/3:C1/1:P4/4:M2/3, meaning that they have three incisors in both the maxilla and mandible (I3/3), one canine (C1/1), and four premolars (P4/4) in each upper and lower hemiarch, and two molars in the maxilla and three molars in the mandible (M2/3). This dental formula is different from a cow or goat, which is I0/3:C0/1:P3/3:M3/3, indicating that they have no incisors or canines in the maxillary arch. Looking at closely related species, the domestic cat has a dental formula of I3/3:C1/1:P3/2:P1/1, while the lynx has a dental formula of I3/3:C1/1:P2/2:M1/1 [52]. Eventually, the number of teeth may vary from the expected dental formula. In these situations, it is important to record which teeth are missing and why, as any deviation in the number of teeth from the dental formula must be considered, such as for example, genetic causes, ante-mortem tooth loss due to disease or trauma, post-mortem tooth loss, or unerupted or undeveloped tooth (agenesia).

Species identification or the distinction of closely related species can also be done using the metric and morphological characteristics of the teeth, applying statistically robust techniques and using advanced tools (e.g., geometric morphometrics) [51,53,54,55,56,57,58,59]. Furthermore, the variation of simple metric characteristics such as tooth size or jaw length can be key in resolving debates about whether a sample comprises a single species or includes more than one morphologically similar species [60]. For example, a high coefficient of variation in a dental sample can be an indication that corresponds to more than one species [61]. However, according to Hillson [59], the absolute size variation of individual teeth is less marked than the relative size variation between different elements of the dentition, so it is important to analyse several classes of teeth at the same time. Of particular importance is the *intercanine distance*, defined as the length between the two tips of the maxillary or mandibular canines (Figure 2). While the shape of the maxillary/mandibular dental arches can help differentiate between mammalian families (it is not possible to distinguish between members of the same family by the shape of their jaws alone), intercanine distance can help differentiate between species of varied sizes in the same family [62] (see Section 3.4.2. Human Deaths from Animal Bites for more information on this topic).

Non-metric dental traits (e.g., presence and size of cusps, form of fissures on occlusal surfaces of premolars and molars, form of ridges, presence of pits) also play a significant role in species identification. Some non-metric traits are normally scored as presence/absence or graded into categories defined by a set of rules [52]. In any case, the variation of these non-metric traits is used to distinguish between species [56]. For example, the pattern of folds exposed on the occlusal surface of persistently equine mandibular cheek teeth varies, this being one of the ways to distinguish different species of horse and donkey [63], although their reliability is currently being questioned [56].

In short, dental form (size + shape) is highly genetically controlled and well reflects phylogenetic relationships, making teeth useful to identify the taxonomy of animals [64]. Thus, species identification is based primarily on macroscopic inspection of dental form (e.g., [65]). In recent years, more complex tools (e.g., geometric morphometrics) and statistical procedures (e.g., machine learning algorithms, artificial intelligence) have allowed to analyse teeth and tooth marks with a higher precision [66,67,68]. However, when teeth are in a poor state of preservation, these traditional or advanced methods could be severely limited due to the difficulty or impossibility of observing species-specific dental anatomical characteristics. In this situation, histomorphometry of dental tissues (i.e., evaluating the organisation, composition, and structural components of enamel, dentine, and cementum) [52], immunological procedures [69], stable isotopes [70], and genetic tools (such as DNA sequencing, Single Nucleotide Polymorphism, Polymerase Chain Reaction–Restriction Fragment Length Polymorphism, and microsatellite analysis) [71,72] have become particularly useful and relatively applicative. However, all these methods require complex, time-consuming, and highly professional procedures. For these reasons, fast, accurate, and easy-to-use methods and techniques have been developed in recent years to identify the species origin from teeth samples. Non-destructive analytical chemistry (i.e., spectroscopy techniques) are constantly evolving and they are widely used in forensic science and practice. Thus, X-ray fluorescence [73,74,75] and Fourier transform infrared spectroscopy [76,77] can provide useful information on the elemental components of teeth to identify the species of animal remains. The main advantages of these spectroscopy techniques are that they provide fast and accurate results and do not require complex analytical procedures.

The investigation of criminal deaths caused by animals has increased considerably in recent decades, so that determining whether the death of a human being was caused by a domestic or wild animal (and its taxonomic classification) is a step of vital importance in forensic settings to determine legal responsibilities [22,62,78,79].

#### 3.1.2. Sex Estimation

Sexual dimorphism is the term that refers to differences between males and females of the same species [80]. This condition is common among mammals, but the levels of dimorphism vary between them, being generally higher in large mammals than in small mammals [52]. Sex is easily indicated by the presence/absence of the *baculum/baubellum* [81], but most frequently sexual dimorphism is identified by body measurements, particularly visible in body mass and size [82]. Sexual dimorphism is strongly present in those species with polygynous social ecology (i.e., a mating system in which one male lives and mates with multiple females but each female only mates with a single male), reflecting increased male–male competition for access to breeding females [83], rather than related to diet, habitat, or activity patterns [84].

Size-related sexual dimorphism is a common phenomenon in carnivores, particularly in the size of the skull, mandible, and teeth, with males on average being significantly larger than females [84,85,86,87,88], except in some animal species such as the spotted hyaena (*Crocuta crocuta*), where a reverse sexual dimorphism is observed [89]. In this order of mammals, sexual dimorphism in the size of the skull, canines, carnassial teeth, and molars is widespread, being more pronounced in the families of Felidae (e.g., [90,91]), Canidae (e.g., [82,92]), and Ursidae (e.g., [93,94]). These taxa contrast with other mammalian groups such as Primates, where diurnal species are generally more sexually dimorphic than nocturnal species, and terrestrial species tend to be more sexually dimorphic than arboreal species [91,95,96].

In general, dental sexual dimorphism of Primates centres on the canines [95,97,98,99,100] and, combined with the rest of the teeth in a discriminant analysis, can be used to assign a sex correctly in skeletal remains. It has been suggested that the sexual dimorphism in the dentition, centred on the canine teeth, is related to the so-called *field effect*, in which teeth closer to the canines tend to be more sexually dimorphic than those further away [101,102,103]. Among Primates, sexual dimorphism in the size of canine teeth ranges from minimal to extreme levels (e.g., lack of sexual dimorphism in *Aotus*, *Callicebus*, *Saguinus*; moderate in *Pan* and *Gorilla* [ca. 25–40%]; extreme in *Papio*, *Mandrillus,* and *Theropithecus* taxa (ca. 69–75%)) [98,104]; data of sexual dimorphism calculated from [105]). Humans fall at the low end of the range of Primate canine dimorphism, with male canine teeth that are up to 10% larger than those of females [98]. Dental sexual dimorphism is also marked in tusks, including marine mammals such as narwhals, walruses, and dugongs, and herbivorous terrestrial mammals such as elephants and hippopotami [52].

Thus, in the resolution of a forensic investigation case, sex estimation is one of the first steps in reconstructing the biological profile of an animal that must be performed. The main reason for this is the fact that other vital information, such as age-at-death and body size, cannot be adequately obtained without prior sex estimation.

#### 3.1.3. Age-at-Death Estimation

In the forensic context, age-at-death estimation (i.e., the amount of time that has passed from the birth to the given date of death) is one of parameters of the biological profile crucial to establish the identity of the animal [106]. The ageing of animals can be particularly important in disputes over purchases [107], control of hunting [108], violation of laws about animal conservation [109], age of abandonment or adoption [110,111], trade and imports [112], and deliberate acts of violence against animals [109], among others.

Age-at-death estimation can be applied to living animals or skeletonised remains [113]. The examination of bones, horns, and dentition has been proposed in ageing of carcasses, as well as the length or height of animal and the colour of the pelage [106]. However, the study of animal dentition is one of the most practical and accurate methods for estimating their age-at-death [114]. Dental age-at-death estimation has been widely accomplished in human forensic investigations and wildlife animal research [111]. The anatomical processes of development occur in the same way in humans and animals [115]; therefore, dental age-at-death estimation methods are focused on changes in growth and development of teeth, as well as the changes after their formation [116]. Several methods have been proposed for the estimation of dental age-at-death in animal forensic investigations, such as those based on (i) dental development and eruption, (ii) occlusal tooth wear, (iii) dental cementum annuli, and (iv) secondary dentine deposition.

Since dental growth and mineralisation follow a consistent sequence and clear-cut changes occur over a brief period, age-at-death can be estimated with reasonable reliability from the state of development [52]. In veterinary practice, age-at-death can be estimated by visual examination evaluating dental eruption, since the sequence and timing of the eruption of teeth provides a reference scale for age-at-death estimation; it can be studied since the tooth begins the process when the crown emerges from the crypt until it reaches the occlusal plane [117]. When estimating age-at-death through dental development, it is necessary to consider the different patterns and types of growth variation within species, taking into account: (i) the basic types of tooth development (i.e., *monophyodont*, *diphyodont,* or *polyphyodont*); (ii) the types of shape of dentition (i.e., *homodont* or *heterodont*); (iii) the anchorage of teeth (i.e., *thecodont*, *acrodont,* or *pleurodont*); (iv) the basic types of tooth crown (i.e., *brachyodont* or *hypsodont*); and (v) the basic types of jaw occlusal overlay (i.e., *isognathus* or *anisognathus*) [118]. For example, in many mammal species the tooth continues to grow after the main eruptive phase, which causes occlusion of the teeth and finishes with the closure of the root apex; for other mammals, the crown or root continues to grow throughout life and, in still other mammals, the root can be completed soon after the initial eruption [52]. The application of this method of age-at-death based on the evaluation of dental eruption is applied in humans only in children or young individuals; this situation must be considered in the case of some animals, for example the elephant, whose molars have an eruption sequence of 30 to 40 years [119]. Descriptive stages for dental development and eruption in relation to chronological age have been proposed in many animals of the order of carnivorous mammals such as canids (e.g., dog; [112,120,121]), felids (e.g., cat; [120]), and non-human Primates (e.g., lemur, galago, chimpanzee, gorilla, macaque; [122,123,124,125]), as well as ungulated mammals such as caprines (e.g., sheep, goat; [120,126,127]), bovids (e.g., cattle; [128]), and suids (e.g., pig; [120,129]), among others.

After the dentition is fully erupted, several researchers have proposed age-at-death estimation methods based on dental wear [117]. Once a tooth emerges from the gingivae, dental wear initiates as a consequence of the grinding of teeth against one another, and the contact with food, cheeks, and tongue [52]. Dental attrition of the permanent teeth has been extensively studied and is considered a classic method for age-at-death estimation in adult animals [119], visually assessing the loss of enamel and the amount of the dentine exposed [52]. The most used protocols for recording dental attrition are based on changes across the occlusal surface of premolars and molars, such as those developed by Payne [130] (a system based on sheep and goat remains) and Grant [131] (for cattle, sheep, goat, and pigs). Since the 1970s, an alternative approach has been proposed based on the measurement of crown height to evaluate its decrease through age (e.g., [132] in zebra, [133] in horse, [134] in red deer). Despite being a widely used method, it presents certain limitations [126], such as: (i) the assumption that there is a degree of correspondence between the tooth wear classes and the chronological age of the animal at death, (ii) the assumption that age ranges are of equal length, and (iii) the discrepancy of applying methods developed in ancient animal populations to contemporary animal populations or vice versa (i.e., to apply an age-at-death estimation method developed in an animal population of a specific chronology to another population of a different chronology).

Another age-at-death estimation method is based on the analysis of incremental structures in dental cementum [117]. The deposition of cementum is continuous throughout the life of the animal, providing a longitudinal record of factors affecting its growth, resulting in incremental bands correlated with seasonal growth in most species [135]. Each annual period of deposition is composed of a ‘summer’ layer (i.e., a wide and translucent layer that corresponds to the growth line) and a ‘winter’ layer (i.e., a narrow and hypermineralised layer that corresponds to the rest line) [136]. When longitudinal tooth sections are observed under a light-transmitting microscope using polarised light, translucent and opaque bands alternate as a result of the growth pattern; so, these bands can be related to the age of the animal and used to conduct the estimation of the age-at-death [135]. However, the regularity of deposition can be influenced by external factors (such as climatic variation or quality of food), but also by internal factors (such as feeding and reproductive habits) [136]. Thus, counting of dental cementum annuli is a very reliable and accurate method to provide estimates of age-at-death for seasonal animals, such as moose [136,137], red deer [136,138], black bear [139], polar bear [140,141], harbour seal [142], ringed seal [143], American badger [144], and feral pig [145], among others.

As in human dentition, the study of secondary dentine deposition inside the pulp chamber is also applied for age-at-death estimation in animals [52]. Secondary dentine is the dental tissue formed after root completion and its deposition is continuous inside the pulp cavity in the form of layers while the pulp remains vital. As a result, the pulp cavity reduces in volume with age [118]. The relationship between the pulp/tooth area ratio using dental radiographic images is the basis of this age-at-death estimation method and has been applied in several animal species such as cat [111], dog [146], coyote [147], and lion [148].

#### 3.1.4. Body Size Estimation

Body size is described in terms of body length or mass, since these two variables provide the greatest predictive value for understanding the animal’s ecology [149]. For example, among living mammals, body size is usually described in terms of mass; however, when the animal is too large to be easily weighed or it is difficult to directly measure this variable in the field, withers height and body length are mainly used (e.g., whales) [150], although it has also been used for other domestic and small mammals [151,152,153].

Limb-based estimations of body mass are the most common methods using either lengths [154,155] and/or midshaft cross-sectional dimensions [154,156] of long bones. They have the advantage that they are based on the relationship between body mass and the load borne by the limbs when they support the body on the ground [149]. However, because teeth are most frequently preserved in the skeletal record, their size is often used to estimate the body mass by biologists and palaeontologists [157,158,159].

For example, current approaches to body size estimation in mammals stem from Gould’s paper [160] in which he proposed that the area of mammalian post-canine occlusal tooth area scales against body mass with positive allometry as a result of metabolism and/or or changes in dietary strategies (i.e., tooth size in mammals scale in a predictable manner to body mass to meet their metabolic needs) [157,161]. While several studies use the post-canine tooth row length to infer allometric relationships with body mass [157,162,163,164], other researchers have proven a strong relationship between body mass and the area of individual teeth, particularly the first molar (e.g., [103,165,166,167,168,169,170,171,172]). The first molar is the most useful tooth for this purpose because it tends to have relatively low levels of intraspecific variation and sexual dimorphism compared with the other tooth classes, probably as a consequence of its early eruption in the oral cavity during ontogeny [165,168,173,174]. Thus, inferring body mass from individual teeth adds to the metabolic scale assumption by assuming a relationship between the size of that tooth and that of the rest of the tooth row [149].

Numerous studies have performed regression equations based on post-canine tooth row length and/or mandibular first molar crown area (i.e., crown area = mesiodistal × buccolingual diameter) and have been developed for a variety of species of the class Mammalia, including ungulates (e.g., [158,159,172]), marsupials (e.g., [168]), carnivores (e.g., [163,171,174]), rodents (e.g., [164,175,176]), Primates (e.g., [157,165,177,178]), and even sharks (e.g., [179]) (Figure 3). Although the first molar is considered the tooth that has the least variation in its adjustment to body mass and, therefore, would be the ideal tooth to estimate body mass from a single tooth, regression equations are available for the other tooth classes of the dentition (e.g., [165,168,174,178]).

For example, in forensic settings, the examination of dental features and characteristics of a bite mark may help identify the animal that caused the biting injury (see Section 3.4. Bite Marks for more information on this topic). Useful clues for classification of the aggressor type include the estimation of the size of the mouth, calculated measuring the intercanine distance. This parameter can help to estimate the animal’s breed and body size and can also help to distinguish between different sized species in the same family [62].

### 3.2. Geographical Origin (Provenance) Identification

An understanding of patterns of movement is fundamental for the knowledge of the ecological, life history, and behaviour of the majority of animals, which requires that specimens be tracked across time and space. Tracking animal movements can be done directly using some type of extrinsic marker (e.g., radio and satellite transmitters) or indirectly using intrinsic natural biological (e.g., fatty acid profiles, DNA techniques) or biogeochemical markers (e.g., concentration of trace elements, stable isotopes) [180,181]. Stable isotope techniques are the most widely used to link an animal to a specific geographical region and are based on the principle ‘you are what you eat’ (i.e., isotopic signatures of foods consumed are recorded in consumer tissues) [182].

Stable isotope ratios vary among biomes that animals inhabit and are incorporated into organism tissues from its diet. In this way, animals moving between isotopically different biomes can retain information of previous feeding locations for periods of time that depend on the turnover rates for the different organism tissues [181]. Keratin-based tissues (e.g., hair, feather, nail, claw, bill) are metabolically inert after synthesis, maintaining an isotopic record reflecting the location where the tissue was synthesised, so they are usually used to study seasonal movements. Conversely, metabolically active tissues’ dietary and source information will correspond to a temporal integration, ranging from a few days or even hours (e.g., urine, faeces, blood plasma) [183,184,185,186] to several weeks (e.g., muscle, whole blood) [187,188,189] or even years (e.g., bone collagen) [190,191,192] (Table 1). Thus, investigations examining long-term movements use metabolically inert tissues, whereas studies on recent movements use metabolically active tissues with rapid turnover rates.

In the case of teeth, stable isotope analysis can be performed on either the organic or inorganic fraction. The organic fraction preserves proteins such as collagen, so the collagen contained in dentine can be used to assess short-term changes that occurred during puppyhood, as these tissues form in early life and undergo little remodelling [193]. The inorganic fraction is primarily formed by hydroxyapatite. Dental enamel, like bone, is mainly composed of hydroxyapatite; nevertheless, unlike bone (a relatively porous material composed of tiny hydroxyapatite crystals interspersed with approximately 30% of organic matter), dental enamel is essentially non-porous, composed of relatively large crystals including only minor amounts (ca. 2% or less) of organic matter [194]. Therefore, the dense crystalline structure of enamel makes it the preferred tissue for isotopic analysis, as it is less susceptible to diagenetic alterations compared to bone tissue [194,195]. Furthermore, dental enamel, unlike bone, is not remodelled during life, and therefore the isotopic signature of dental enamel is directly related to the environment and diet during the period of tooth formation [195].

In recent decades there has been an increase in the use of stable isotope analysis in wildlife and criminal forensic contexts as a means of determining the origin and movement of animals [196,197,198,199], with applications as varied as establishing the origins of ivory from elephant tusks (e.g., [200,201,202,203]), or the illegal animal trade or animal derivatives. An example of the latter case is the increase in seizures of jaguar (*Panthera onca*) body parts (e.g., skin, fat, meat, claws, tails, skulls, bones, teeth), which have been occurring since 2013 in different Latin American countries. This increase in seizures is linked to the high demand in Chinese markets for body parts of these felines to satisfy the demands of traditional Asian medicine [204,205]. In the particular case of teeth, these are used locally for decorative (e.g., jewellery and key chains), medicinal (e.g., therapeutic treatment for facial paralysis caused by a spell of misfortune; dental fillings), or for cultural purposes (e.g., necklaces used in traditional festivals; amulets to protect against bad luck or evil spirits) [206]. The application of stable isotope analysis in this context is considered a useful tool in countering wildlife trafficking efforts [207].

### 3.3. Post-Mortem Interval Estimation

The post-mortem interval (also known as the *time since death*) is the time between the death of an animal and the discovery of the body [208]. Knowing the time since death is essential in the investigation of human deaths, as well as in animal forensic investigations [209,210]. Various reasons have been proposed for estimating the post-mortem interval in animals: (i) inclusion or exclusion of individuals from a group of suspects and corroborating witness testimony [209]; (ii) differentiation of single or continuous episodes with animals [208,211]; (iii) identification of incidents of hunting out of season, poaching, negligent transport of animals or abuse of companion animals [208,211]; and (iv) application of legal deadlines (e.g., disposal of animal carcasses) [208,211]. In human forensic medicine, the study of the post-mortem interval is one of the most popular topics; however, in veterinary forensics, the number of studies is extremely limited [208,209,210,211]. Researchers must face a deficiency in the development of methodologies for a large number of species and, therefore, the obligation to apply methods developed in humans, lacking the appropriate validation to be applied in crimes against animals [209,210,212].

The most used methods of relevance to forensic veterinary pathology for estimating the post-mortem interval in animals’ dead bodies are mainly based on temperature changes, muscular stiffening (also called *rigor mortis*), ocular changes, cadaveric lividity (*livor mortis*), decomposition processes, and entomology [210,212,213,214]. In the case of studies conducted on animal dentition, there is a limited amount of research based on morphological, histological, or molecular analysis [215]. For example, Akbulut et al. [216] analysed the relationship between the changes in the mineral density of enamel, the surface abrasion of hard dental tissues, and the estimation of time of death through micro-CT. The results of this study realised with rats’ dentition showed that morphological changes in the microstructure of dental tissue can be considered a parameter for estimating the post-mortem interval in a forensic context. In the same way, a study by Granrud and Dabb [217], based on the exfoliation of the anterior dentition of pig teeth, showed a potential relationship with the post-mortem interval. Thus, an accumulated degree-days can be used to establish a minimum length of time since death; however, the authors indicated that future research is necessary. Mehendiratta et al. [218], on their part, observed a series of morphological and histological changes by analysing the dental pulp, showing that this approach could be applied in the early phases since after 144 h, dental pulp is not preserved (i.e., it is completely decomposed). Finally, Young et al. [219], analysing the dentition of buried pigs, studied the progressive changes in pulp coloration due to decomposition and the post-mortem pulp RNA degradation. The morphological and molecular evaluation showed a favourable estimation of the post-mortem interval through the differences in RNA decay and progressive colour changes. The authors highlighted the advantages of this method for its speed of application and low cost.

The small number and the results of the studies conducted on animal dentition for estimation of the post-mortem interval show the need to increase the analysis on this topic. All the studies point out the potential of their methods but emphasise the need for further research to give greater solidity to the results [216,217,218,219]. In the same way, they indicate the importance of developing studies on taphonomic factors to improve the precision of the post-mortem interval estimation. The choice of the study animal (mostly pig) represents the objective of application in human forensic cases. The majority choice of the pig is probably due to its similarity as an animal proxy to the human and not because of the need to study it as the animal victim or perpetrator. Similarly, the introduction and discussion of the studies reviewed are contextualised within human forensic activity and not in the animal context.

### 3.4. Bite Marks

In forensic sciences, recognising and correctly identifying the actions of animals on human remains, but also on other faunal remains, is crucial, as this allows the collection of data about events that may have affected the body over a time, which may have ranged from the ante-mortem to post-mortem period [220]. In certain contexts or situations, animals can cause severe injuries that, on one hand, may lead to the death of the individual attacked and, on the other hand, can alter the corpse in the post-mortem period, either in relation to soft or hard tissues [221]. To reconstruct the forensic scene as reliably as possible and define how certain animal species acted on a human body, it is essential to correctly identify the nature of the injuries, the anatomical region affected, the circumstances in which they occurred and the agent that caused them, in order to avoid possible misjudgements with very disastrous implications in the forensic framework. At a macroscopic level, bite marks are among the signs most frequently found on the body of a victim, whether it is exposed in an open, outdoor, or enclosed environment. Bite marks can be defined as both superficial and deep marks left by teeth that affect, in diverse ways, both soft and hard tissues whose morphology varies depending on the size and shape of the maxillary/mandibular dental arches and the force exerted by the bite [222]. According to Binford’s standards [223], four basic types of tooth marking can be recognised by the motion of animal teeth over the surface of bones: (i) punctures, (ii) pits, (iii) scores, and (iv) furrows.

Punctures are those regular-shaped marks left clearly and evidently at the point where the bone has collapsed under the pressure exercised by the teeth (e.g., canines). Since perforations are a direct consequence of the force that the animal’s bite has exerted on the bone, the specific types of shapes and sizes will depend directly on the species involved (e.g., large and deep tooth marks could be due to the action of large carnivores such as canids or felids). However, puncture marks are not always circumscribed and well-defined and, in some cases, it is possible to find puncture marks associated with gnawing marks on the bone after the consumption of the soft tissue [223].

Gnawing generally proceeds from soft to hard bone; the animal first attacks the soft spongy parts of the bone and only later encounters progressively harder bone (i.e., compact bone), where pitting can occur. That is, the same actions of gnawing may result in pitting since the compact bone is strong enough to not collapse under the action of gnawing. According to Binford [223], when pitting is present rather than puncture marks, this indicates a prolonged gnawing action on the bone that cannot be attributed to either consumption or extraction of the meat from the skeleton, as is the case of killing.

Scoring is the result of either turning the bone against the teeth or dragging the teeth across relatively compact bone. The result is a scarring of the surface with close, linear morphological characteristics. In the case of tooth scoring, it is important to examine their characteristics carefully so as not to make the mistake of interpreting them as marks caused by different classes of bladed weapons [223] (Figure 4).

#### 3.4.1. Animal Scavenging

Animal scavenging is one of the main post-mortem taphonomic processes responsible for the modification of a corpse, thus having a significant implication in forensic casework, especially in cases where the remains are deposited in an outdoor environment [225,226]. Identifying the nature of injuries made by domestic and/or wild animals on human bodies in which soft tissue is still present at the time of scavenging or on already skeletonised remains is still quite complex. In some cases, these can be misinterpreted as traumatic injuries caused by sharp weapons and gunshots, with signs of interpersonal violence that occurred ante-mortem or peri-mortem and with additional taphonomic variables [227]. Since the bite marks left by scavenging animals can lead to difficulties in the interpretation of the forensic investigation and consequently, also compromise the identification of a perpetrator and the interpretation of the cause of death, it is crucial to be able to recognise them, considering how the nature of the bite marks vary depending on the species involved, how behavioural patterns of predation vary at local and regional levels, as well as the dispersion or alteration of human remains induced by these scavengers [228,229,230].

Generally, depending on the animal species involved in scavenging, both superficial injuries affecting only soft tissue and situations of disarticulation, dismemberment, and dispersal of body parts can be recorded, further complicating the recovery process and identification of human remains [231]. Therefore, knowledge of the regional fauna by veterinarians, biologists and forensic odontologists not only provides a better understanding of which animals feed on the carrion, but also of the reasons why they consume corpses. The involvement of these categories of experts in forensic investigations is proving to be increasingly necessary in the field of forensic activities since in complex situations it prevent errors when investigating bite marks caused by a possible animal attack on humans. 

Although sometimes it is complex to identify the scavenger species that acted directly on the body, specialists can make use of some standard procedures beginning with macroscopic and direct observation of visible marks. This is followed by an assessment of the environment and geographical context of the finds, the pattern of scavenging on a body or skeletal element, the type and size of teeth marks, and finally, the additional faunal evidence associated with the forensic scene such as excrement, hides, or other organic elements. In more complex cases, molecular genetic analyses may also be performed to identify or exclude the perpetrator of certain injuries [232]. Therefore, in addition to the types of teeth marks, it is equally important to be aware of the different species of scavengers present around the forensic investigation as each of these leaves a different bite mark on soft tissues and bones. The main groups of scavengers that leave visible traces on human bones include (i) carnivores (e.g., canids, felids, and ursids), (ii) ungulates, (iii) rodents, and to a lesser extent (iv) birds. The mechanism of scavenging and the marks left on the soft tissues and bones between these species are quite different [233]. Large canids and felids (e.g., dogs, foxes, coyotes, wolves, lions, lynxes, cougars) are among the major carnivores on the planet capable of attacking or killing a human being, as well as being responsible for scavenging. In cases of displacement, it is important for forensic investigators to understand the behavioural patterns and patterns of animals that move locally; in fact, dismemberment and dispersion may occur differently depending on the species, but also on climate, season, and body size of the animal [234]. These variables must always be considered in a forensic investigation, as knowing how the local fauna acts in certain situations and how it approaches the remains are crucial, especially when recovering remains that are scattered over an area that may vary from a few to tens of metres [235].

Although the modalities and patterns of animal scavenging vary depending on the species involved, the behavioural pattern of predation and the availability of access to the body (e.g., open/enclosed environment, clothing, concealment by vegetation), the main cause of scavenging is the consumption of soft tissue and bone for nutritional purposes [236], which in some cases may be scarce in the ecosystems in which they live [237]. From a behavioural point of view, large carnivores first consume the soft tissues of a body from the thoracic cavity and only later turn their attention to the limb bones, leaving clear signs that can be traced back to both the action determined by claws and teeth [238]. Generally, the bite marks of canids and felids (*puncture marks*; see Section 3.4. Bite Marks) are recognisable in comparison to those of other animal species because the canines of these two families gnaw soft tissue or chew long bones to extract meat, fat, and bone marrow, leaving typical lesions in the form of pits and ovoid holes [239]. Like carnivores, rodents are also often found interacting with human remains in forensic contexts, leading in some cases to the dispersal of small bones, such as those of the hands and feet [240,241]. Rodent activity can be recognised by the presence of specific imprints known as *gnawing marks* ([223]; see Section 3.4. Bite Marks). Rodent bites are typically found post-mortem; these produce osseous changes through the gnawing action exerted by the upper and lower incisors constantly moving over skeletal remains (Figure 5). Unlike the bite marks of canids, those of rodents are distinguishable in that they do not usually affect soft tissue or leave claw marks but are mostly concentrated around the diaphysis of bones [226]; bone tissue, in addition to being a food source of calcium, is also an efficient wear tool for rodents to file away the continuing growth of incisors. Although not as significant, some species of herbivores, including ungulates (e.g., cervids), also constitute a taphonomic agent regarding the modification of human remains; these, in fact, in an open environment may proceed to consume bones due to a lack of phosphorus in nature. Teething, in this case, leaves a visible imprint in the form of erosion or exposure of the spongy tissue of the bone, sometimes associated with a series of parallel grooves smaller in size than those left by rodents. In long bones, the chewing of proximal or distal epiphyses by ungulates results in the formation of impressions whose morphology is termed *fork-shaped* [242,243,244].

#### 3.4.2. Human Deaths from Animal Bites

In forensic contexts, there are frequent cases of death resulting from a violent predatory attack by certain animal species. Compared to the number of attacks by domestic animals on humans, those involving wild animals are much rarer and usually involve deceased humans [245] (see Section 3.4.1. Animal Scavenging). However, when tragic events of this type occur, there are several factors to take into consideration to understand the circumstances of the attack, the identity of the animal involved, the behavioural pattern in terms of both predation and feeding and finally, the consequences for the victim and their ante-mortem reactions [246]. In this way, in the case of a victim with physical evidence on the body of direct involvement of an animal predator, careful documentation and analysis of the anatomical region fatally involved, the position of the victim’s body at the time of discovery, the age of the victim and the level of violence of the attack need to be noted; the size and morphology of the dentition involved, especially when there are clearly visible bite marks in soft tissue and bone, is helpful in correctly identifying the animal species responsible for the attack [247]. The analysis of morphological and metric characteristics of bite marks by veterinarians, biologists, and forensic odontologists can be particularly useful when trying to determine or discard the species, size, sex, and age of the biter involved in the attack [248]. Among domestic animals, dogs, especially in group situations, are the most frequently responsible for aggression against humans with often fatal consequences [249,250]. Injuries caused by a bite can vary from mild with non-fatal consequences, to profound with the consequent death of the victim, also including deep post-mortem lacerations and tears of soft tissue of the victim’s body, caused by the action of canine teeth [251,252,253].

Thus, the analysis of bite marks in forensic contexts may provide sufficient scientific evidence that would help identify or exclude the animal perpetrator. This is especially useful when different animal species are involved in the area where the corpse was found, whose behavioural and injury patterns on the victim’s body may be similar and whose misinterpretation may lead to errors and incorrect interpretations in the determination of the cause of death and the ante-mortem activities of the animal, often with serious consequences in the resolution of legal cases. An interesting case in this regard is the case report published by Fonseca and Palacios [22] concerning a case of animal predation on a male victim found in Argentina. The two experts, one an expert in forensic odontology and the other in animal biology, were involved by the competent judicial authorities to determine the nature of the injuries found on the victim’s remains and to establish the animal species involved in the attack that turned out to be fatal for the victim. The multidisciplinary investigation aimed to obtain additional information on the manner of the attack and the animal species responsible for the fatal injuries, since the initial hypothesis concerned both the possibility that the attack was carried out by a cougar (*Puma concolor*) or by a pack of large dogs roaming the area where the victim’s body was found. The main problem with the correct identification of the perpetrator of the death was the size and characteristics of the injuries found on the corpse, which are similar for both species. Following a careful assessment of the type of wounds, the anatomical position of the lesions, the tooth marks, the behavioural pattern of both animal species present in the environment, together with information of several eyewitnesses regarding previous aggressive behaviour of a pack of mixed-breed dogs of various sizes (e.g., Dogo Argentino, English Mastiff) found in the same area, the hypothesis regarding the dog pack being responsible for the attack and death of the victim was confirmed.

Another interesting situation involves canids. Wolves are considered a large and broadly distributed population in Europe. They preferably prey on wild animals and domestic ungulates (e.g., deer, cattle), causing great conflicts with humans and their economic interests in the case of livestock. Dogs are also widely distributed in Europe, coexisting with wolves, which means that wolf predations and bite marks sometimes can be confused with dog attacks [79,251]. There are plenty of cases in which the physical and circumstantial evidence at a crime scene refers to a fatal attack perpetrated against humans by wolves [23]. Wolves, however, are only under specific conditions responsible for violent attacks on humans, but like other large carnivores they participate in post-mortem scavenging. In this regard, a study by Toledo-González et al. [59] on the Iberian wolf and its associated bite marks to distinguish them from other canid species is interesting. The researchers described dental morphometric characteristics to aid the identification of the Iberian wolf’s tooth/bite marks or to rule out other potential aggressors with great confidence.

Determining which predator species is responsible for killing a human is important, especially when there is the possibility of overlapping bite marks, as is the case with many carnivore species [62] (Table 2). In bite mark comparisons of sympatric animals (Note: *sympatry* is the term used to describe populations, varieties or species that occur in the same place at the same time [254]), measurements of the maxillary and mandibular intercanine distance are frequently used as an aid in identifying the different animal species responsible for a predatory or scavenging attack [255,256] (see Section 3.1.1. Species Identification for more information on this topic). The exclusion of one species over another in forensic contexts involving human remains, as well as in situations involving the interaction of wildlife, livestock, and humans, is crucial for determining appropriate management actions and for avoiding inappropriate human actions, such as the unjustified killing of certain predator species, mistakenly believed to be responsible for deadly and endangered attacks [79]. In relation to attacks by certain species on domestic livestock, Verzuh et al. [257] recently examined the mandibular and maxillary intercanine distance of Mexican wolves and sympatric carnivores (i.e., grey foxes, bobcats, coyotes, feral dogs, and cougars) in order to identify the measurements of intercanine distance that had the most potential to be multiple species and therefore, the least reliable measurements for bite mark analysis. The results of this study showed how indeed many of the measurements examined overlapped with each other, thus generating confusion in identification. However, it may also be useful for forensic investigators to know the measurement intervals considered problematic so that, together with the rest of the information obtained from a case, they can proceed with caution in the judicial investigation or at least not draw risky conclusions about the species involved in the predation.

## 4. Application Potentialities in Other Contexts

So far, we have seen the importance of dentition in the identification process in forensic contexts. However, in addition to having an interest in the legal system, the analysis of animal dentition also has practical applications in other contexts where it is important to reliably identify the finding of animal remains, especially when they have suffered taphonomic alterations (e.g., fragmentation, cremation, cortical erosion) and teeth are the only available elements for analysis.

### 4.1. Archaeological and Palaeoanthropological Contexts

The analysis of bone modifications is the object of study in important disciplines such as bioarchaeology, palaeoanthropology, and zooarchaeology since, on the one hand, it allows understanding of the factors that caused such modifications, and on the other hand, allows reconstruction of the processes that led to the dislocation, fragmentation, and/or poor state of preservation of certain skeletal elements in the funerary context [260]. After the burial of an individual several physical, chemical, and biotic agents (e.g., fossorial species such as badgers, moles, voles, shrews, earthworms, beetles, ants; terrestrial species such as wild cats, lynxes, red foxes, raccoon dogs, brown bears, wild boars, squirrels; Note: *fossorial species* are those adapted to burrow into the ground and live primarily, but not solely, underground [261]) affect the body of the deceased. These agents are responsible for certain degradation processes that manifest themselves in the form of heterogeneous changes visible on the osteological remains (see Section 3.4. Bite Marks). It is necessary to take the utmost care when observing and analysing these macroscopic marks because in bioarchaeological and palaeoanthropological contexts it is common to find human remains that show clear traces of modifications related to animal action, but which, however, can easily be confused with the effect of pathological lesions or due to an incorrect excavation, transport, or storage of the remains by unqualified workers (Figure 6). The erroneous interpretation of the observed bone lesions can result in the loss of useful information for the interpretation of the post-depositional archaeological context and for further anthropological analyses. Among the bone lesions found most frequently in bioarchaeological contexts, those associated with the gnawing action of rodents or carnivores stand out [262], while in the palaeoanthropological context they are associated with the practice of scavenging by large carnivores [263]. In both cases, these bony lesions inform the researcher about the history of taphonomic damage after the death of an individual and before excavation/recovery.

The extent of osteological damage in archaeological contexts varies depending on the animal species involved, the environmental conditions, and the nature of the archaeological context (e.g., necropolis, isolated burials, caves) [264]. The attention of researchers to the carnivore tooth marks found on the skeletal remains of palaeoanthropological assemblages belonging to hominins is justified by the desire to understand the relationships and ways of living and sharing the same environments between large carnivores and humans; especially in prehistoric times, where some known examples of hominin remains show traces of bone modifications that could be associated with forms of feeding by certain animal species [265,266]. The possibility of identifying specific carnivores from the marks of their teeth left on hominin bones could provide sufficient evidence to demonstrate their presence in archaeological, palaeoanthropological and palaeoenvironmental times even when their skeletal remains are not preserved. Moreover, being able to distinguish which carnivores were active in the sites where faunal and human skeletal assemblages are recorded could provide useful information to reconstruct palaeoanthropological scenarios and to understand, if there were, patterns of competition between hominines and carnivores to access carcasses for nutritional purposes [267,268]. Furthermore, the interest in teeth marks left by carnivores on skeletal remains of hominines is justified by the fact that it is considered much more likely to find bite marks left by large carnivores on the bones of hominines than the other way around [266].

Interactions between hominins and large carnivores have occurred with great frequency and in different forms throughout the course of evolution, so much so that in many cases they have generated a series of reciprocal pressures [269,270,271]. From these interactions, the scenarios that generally seem to emerge include violent human–carnivore conflicts (associated with bone modifications), competition for the use of living habitats such as caves, exploitation, and scavenging of common prey or carcasses by large carnivores and hominins.

Examples of human remains associated with animal scavenging activity are particularly recorded during the Pleistocene; in fact, several archaeological assemblages relating to both the accumulation of animal bones and hominins bearing signs of chewing by carnivores have been found at many sites from this period. The fact that prehistoric hominin remains show clear signs of animal biting is important evidence in palaeoanthropology and archaeology of the consumption of human remains by carnivores; however, it is not always possible to trace with absolute certainty the animal species directly involved in the scavenging and chewing of such remains [272].

Among the problematic issues related to the identification of both the type of bite mark and the animal species responsible for bone modification in archaeological contexts there is the interference of multiple physical and biological taphonomic agents, which may have succeeded one another and modified the initial context in a time interval ranging from the short to the long term. Another issue is the possible recurrence of events related to the chewing of the same remains by animals distinct from those responsible for the primary consumption event. In this regard, if traces of modification associated with different animal species are found on the same remains, a further problem concerns the possibility of establishing the sequence of the consumption action as well as understanding whether it is a predation phenomenon or secondary consumption of the carcass [273].

There are few cases in which the animal species involved has been identified with certainty or in which a relative hypothesis has been formulated. Interesting in this regard is the study by Daujeard et al. [274] on the remains of a Pleistocene human femur from North Africa with traces of teeth marks interpreted as resulting from the action of chewing for food by a large mammal, more probably a hyena. As in forensic cases, in palaeoanthropological contexts, bite marks have been recorded on fossil hominin remains applying the method of intercanine distance to identify the animal species involved in the formation of the tooth marks. An illustrative case is given by the analyses carried out on the remains of a skull belonging to a juvenile hominid from Swartkrans, South Africa. The analysis of the intercanine distance, together with the morphological study of the signs left by the dentition, led to the hypothesis that we were dealing with an attack by a leopard on a hominin [267,275].

### 4.2. Ecological Contexts: Study of the Biodiversity of Wild Animal Species

The identification techniques that are frequently employed in the legal system related to criminalistics can be applied in other contexts, where the identification of recovered osteological remains is also required, such as in ecological contexts. The study of wildlife remains is also important when unidentified carcasses are reported for monitoring and census purposes in parks and nature reserves in terms of health checks, hypotheses about the cause of death (e.g., investment or poaching), or the impact on biodiversity (e.g., habitat destruction, introduction of invasive non-native species, over-exploitation of resources, climate change). Censuses of various animal species, especially those considered at risk, have been conducted for many years in protected areas; in fact, the application of extrinsic marks (e.g., radio and satellite transmitters) on the wildlife in these areas is fundamental since it also allows the collection of additional information on the use of certain ecosystems. Reports of dead wildlife with extrinsic markings or animals that lack them are also of great interest, since their identification favours the collection of information that is useful not only for a better understanding of natural ecological phenomena, for planning actions aimed at protecting certain species, but also for assessing the risk of the spread of certain diseases transmissible to wildlife [276].

## 5. Concluding Remarks

There is no doubt that veterinary forensics is becoming increasingly important in our modern society, increasing the demand for investigations related to crimes against animals or investigations of criminal deaths of human beings involving animals. The high degree of mineralisation of dental tissues results in their hardness, durability, and resistance to post-mortem insults, so frequently the teeth are well preserved relative to bone tissue when an animal body is recovered a long time after death. Although the identification of carcasses in veterinary forensics is of less importance compared to its counterpart in human forensic medicine, the post-mortem dental profile of an animal can contribute essential information in resolving legal disputes involving animals and animal derivatives, such as species identification, sex, age-at-death, body size, geographical origin, and post-mortem interval. Moreover, the examination of dental features and characteristics of a bite mark may help identify the animal responsible for aggression against humans with often fatal consequences. This review points out the potential of dentition in the identification process in forensic contexts and emphasises the need for further research to give greater solidity to the results, helping the Courts in answering questions of interest to the legal system to reach a reliable verdict.

## Figures and Tables

**Figure 1 animals-12-02038-f001:**
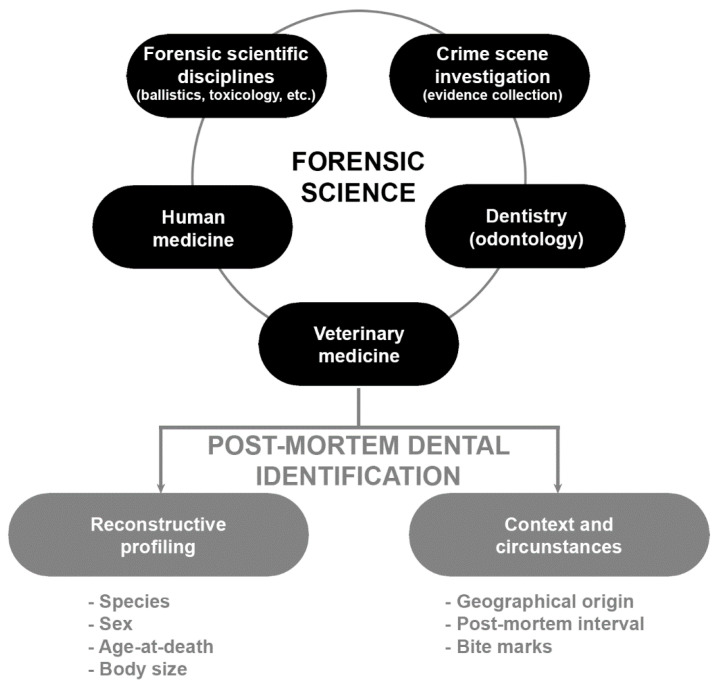
Diagram of the integration of veterinary medicine within the forensic sciences summarizing the main applications of dental profile in veterinary forensics.

**Figure 2 animals-12-02038-f002:**
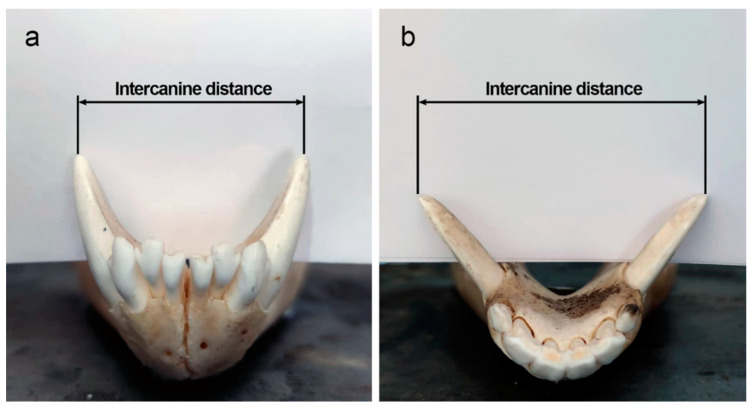
Intercanine distance as measured at the mandibular canine cusp tips in (**a**) *Canis lupus* and (**b**) *Sus scrofa*. Images courtesy of C. Tanga.

**Figure 3 animals-12-02038-f003:**
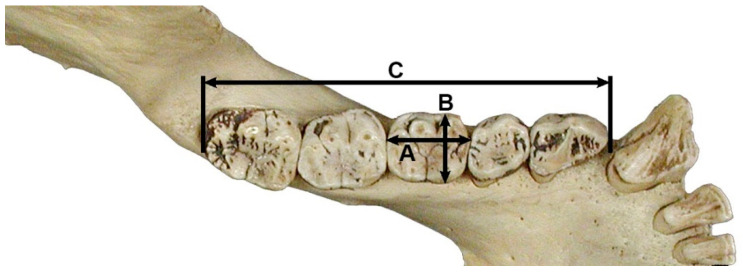
Example of three dental dimensions commonly used for estimating body mass using regression methods, illustrated on the left hemimandible of *Gorilla gorilla*. A = mesiodistal diameter of first molar; B = buccolingual diameter of first molar; C = mandibular post-canine tooth row length.

**Figure 4 animals-12-02038-f004:**
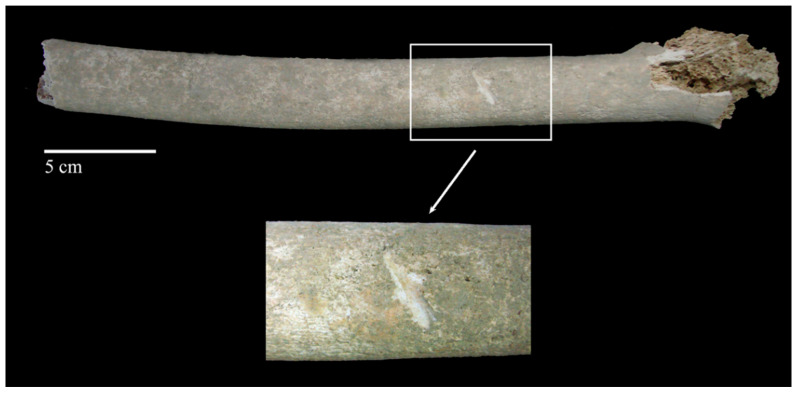
Human skeletal remains from the late medieval cemetery of Corfinio (12th–13th century CE, Italy). Anterior view of the diaphysis of the left femur from an adult individual showing a large linear mark that could be confused with a mark caused by a canine tooth dragging across compact bone; however, this large mark was caused by a sharp tool during a deficient excavation process (i.e., lesion of post-mortem origin). Image courtesy of C. Tanga from her bachelor’s thesis [224].

**Figure 5 animals-12-02038-f005:**
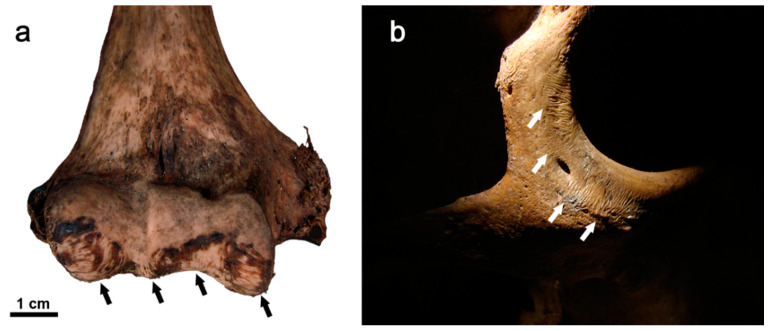
Rodent modifications on human bones in the form of parallel striations from a forensic casework (indicated by arrows). Rodent gnawing signs of post-mortem origin (**a**) on the distal epiphysis of a right humerus and (**b**) on infraorbital and lateral margins of the right zygomatic bone. Images courtesy of J. Viciano.

**Figure 6 animals-12-02038-f006:**
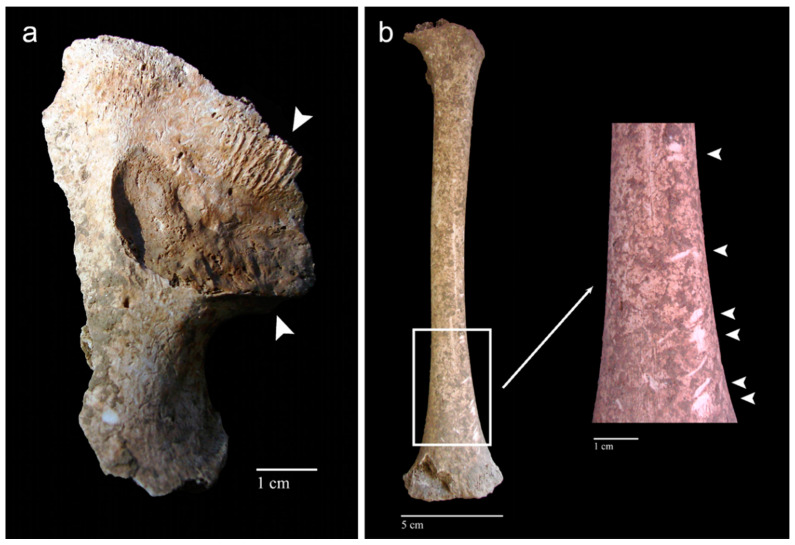
Human skeletal remains from the late medieval cemetery of Corfinio (12th–13th century CE, Italy). (**a**) Fragment of right coxal from a juvenile individual showing signs of rodent gnawing activity near the auricular surface (arrowheads). (**b**) Anterior view of the left femur belonging to a juvenile individual showing scratches that can be confused with signs of tooth marks; however, they are caused by a deficient excavation process using inadequate tools (i.e., lesions of post-mortem origin) (arrowheads). Images courtesy of C. Tanga from her bachelor’s thesis [224].

**Table 1 animals-12-02038-t001:** Time periods represented by isotopic ratios in different body fluids and soft and hard tissues of animals.

Body Fluids and Tissues	Time Period Represented				
	Hours	Days	Weeks	Months	Years
Urine	✓	✓			
Faeces	✓	✓			
Blood plasma	✓	✓			
Blood cells		✓	✓		
Hair		✓	✓	✓	✓ *
Nail		✓	✓	✓	✓ *
Claw		✓	✓	✓	✓ *
Bill		✓	✓	✓	
Feather		✓	✓	✓	
Antler		✓	✓	✓	
Muscle		✓	✓	✓	
Bone (collagen)					✓
Tooth (dentine, enamel)					✓

* Depending on their lengths, hair, nails, and claws can record incremental records of several years.

**Table 2 animals-12-02038-t002:** Ranges of intercanine distances in some species of the order Carnivora (data from [62,256,257,258,259].

Family	Species	Common Name	Intercanine Distance (mm)	
			Maxilla	Mandible
Felidae	*Felis catus*	Domestic cat	7.0–22.0	4.0–18.0
	*Felis catus*	Feral cat	8.2–21.0 *	
	*Lynx canadensis*	Canada lynx	22.7–27.6	21.9–27.8
	*Lynx lynx*	Eurasian lynx	13.0–30.0	5.0–25.0
	*Lynx rufus*	Bobcat	11.0–31.0	5.0–24.0
	*Puma concolor*	Cougar	21.0–48.0	10.0–44.0
Canidae	*Canis latrans*	Coyote	15.0–39.1	7.0–39.0
	*Canis lupus*	Grey wolf	23.0–51.0	11.0–45.0
	*Canis lupus baileyi*	Mexican wolf	31.4–49.8	27.8–43.7
	*Canis lupus familiaris*	Domestic dog	13.0–48.0	6.0–49.0
	*Canis lupus familiaris*	Feral dog	14.1–53.4	10.8–46.2
	*Urocyon cinereoargenteus*	Gray fox	9.0–22.0	4.0–20.0
	*Vulpes lagopus*	Arctic fox	19.4–24.2	16.4–20.8
	*Vulpes macrotis*	Kit fox	13.8–17.5	11.3–15.5
	*Vulpes velox*	Swift fox	15.0–17.0	13.6–15.2
	*Vulpes vulpes*	Red fox	11.0–27.0	4.0–25.0
Mephitidae	*Mephitis mephitis*	Striped skunk	12.0–16.4	10.3–14.8
	*Spilogale gracilis*	Western spotted skunk	8.5–11.6	7.5–10.5
Mustelidae	*Genetta genetta*	Spotted genet	9.0–13.0 *	
	*Gulo gulo*	Wolverine	18.0–43.0	7.0–32.0
	*Lontra canadensis*	Northern river otter	16.7–22.6	14.6–21.4
	*Martes americanus*	American marten	8.6–14.6	8.2–12.5
	*Martes foina*	Stone marten	11.0–15.0 *	
	*Mustela erminea*	Stoat	3.5–7.1	2.6–5.1
	*Mustela frenata*	Long-tailed weasel	5.8–9.0	4.6–8.0
	*Mustela furo*	Ferret	7.8–13.6 *	
	*Mustela lutreola*	European mink	8.0–12.0 *	
	*Mustela nivalis*	Least weasel	3.2–5.3	2.7–4.4
	*Mustela putorius*	Polecat	8.0–14.0 *	
	*Neogale vison*	American mink	8.3–12.4	6.2–10.1
	*Taxidea taxus*	American badger	24.8–33.9	23.8–32.3
Phocidae	*Phoca vitulina*	Harbour seal	27.5–36.3	23.0–36.1
Procyonidae	*Bassariscus astutus*	Ringtail	9.5–12.2	8.6–11.8
	*Nasua narica*	White-nosed coati	19.1–29.3	16.4–22.1
	*Procyon lotor*	Raccoon	19.4–28.2	16.1–25.6
Ursidae	*Ursus americanus*	Black bear	20.0–64.0	11.0–52.0
	*Ursus arctos horribilis*	Grizzly bear	34.0–96.0	15.0–91.0

* Maxillary and mandibular intercanine distances are combined.

## Data Availability

Not applicable.

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
