# Peer review of "Post-Mortem Dental Profile as a Powerful Tool in Animal Forensic Investigations—A Review"

_animals, 2022, doi:10.3390/ani12162038_

Round 1
Reviewer 1 Report
The authors point out the importance of the veterinary sciences in forensic field. This approach ennobles all the knowledge of animals (anatomy, physiology, pathology ...) from legal point of view. This manuscript frames animals as living beings worthy of rights. I think that it deserves to be published.
Author Response
We would like to thank the Academic Editor and the Reviewers for the relevant suggestions that have enabled us to improve our manuscript, entitled “Post-mortem Dental Profile as a Powerful Tool in Animal Forensic Investigations—A Review”.
We have carried out the full revisions requested, and we hope that you and the Reviewers will now find our manuscript of sufficient quality and reader interest for publication in Animals. Please find below the full listing of Reviewer comments and indications of the revisions/ modifications that are now incorporated into our revised manuscript. The changes are highlighted in the manuscript using Microsoft Word’s ‘track changes’ editing feature.
COMMENTS TO REVIEWER 1
The authors point out the importance of the veterinary sciences in forensic field. This approach ennobles all the knowledge of animals (anatomy, physiology, pathology ...) from legal point of view. This manuscript frames animals as living beings worthy of rights. I think that it deserves to be published.
Reply: Thank you very much for your comments, we really appreciate them.

Reviewer 2 Report
This review deals with a very interesting and an important veterinary forensic topic. However, there some concerns need to be addressed as follows:
1. The authors did their best to illustrate all the points in much detail but this makes the review seems like a book chapter. Hence, to edit this manuscript to be in the form of a review article, the authors should remove all general information and definitions that present an introduction to the specific topic of the review. E.g. lines 43-50, the definition of forensic science should be removed. Also, on lines 505-529, the definition and importance of forensic taphonomy of post-mortem interval estimation should be removed and the authors should begin directly from 530. These were just examples but the authors should revise all sections of the review as previously mentioned.
2. Throughout the manuscript, the authors should shorten the long paragraphs. E.g. lines 33-37.
3. Keywords: “geographical origin (provenance) identification” is not suitable to be a keyword. Please, revise.
4. The data in Tables 1 and 2 are not suitable to be presented in tables. The information within these tables should be presented as paragraphs.
5. The authors are highly recommended to add a schematic figure summarizing applications of dental profile in veterinary forensics.
Author Response
We would like to thank the Academic Editor and the Reviewers for the relevant suggestions that have enabled us to improve our manuscript, entitled “Post-mortem Dental Profile as a Powerful Tool in Animal Forensic Investigations—A Review”.
We have carried out the full revisions requested, and we hope that you and the Reviewers will now find our manuscript of sufficient quality and reader interest for publication in Animals. Please find below the full listing of Reviewer comments and indications of the revisions/ modifications that are now incorporated into our revised manuscript. The changes are highlighted in the manuscript using Microsoft Word’s ‘track changes’ editing feature.
COMMENTS TO REVIEWER 2
This review deals with a very interesting and an important veterinary forensic topic. However, there some concerns need to be addressed as follows:
- The authors did their best to illustrate all the points in much detail but this makes the review seems like a book chapter. Hence, to edit this manuscript to be in the form of a review article, the authors should remove all general information and definitions that present an introduction to the specific topic of the review. E.g. lines 43-50, the definition of forensic science should be removed. Also, on lines 505-529, the definition and importance of forensic taphonomy of post-mortem interval estimation should be removed and the authors should begin directly from 530. These were just examples but the authors should revise all sections of the review as previously mentioned.
Reply: We agree that some parts of the manuscript can be shortened. However, we believe that this manuscript is aimed at researchers without extensive knowledge in forensic sciences, so we have seen fit to keep some parts for more clarification of different topics.
- Throughout the manuscript, the authors should shorten the long paragraphs. E.g. lines 33-37.
Reply: This paragraph was shorted.
- Keywords: “geographical origin (provenance) identification” is not suitable to be a keyword. Please, revise.
Reply: The keyword was corrected.
- The data in Tables 1 and 2 are not suitable to be presented in tables. The information within these tables should be presented as paragraphs.
Reply: The information of Tables 1 and 2 were removed and presented as paragraphs. The rest of the Tables were renumbered according to this change.
- The authors are highly recommended to add a schematic figure summarizing applications of dental profile in veterinary forensics.
Reply: A new Figure (Figure 1) was added to the manuscript. The rest of the Figures were renumbered according to this change.

Reviewer 3 Report
The work is interesting and addresses a relatively rarely addressed issue.
[80] Injuries may be due to cultural conditions determining the manner of killing.
[255] Please add that the presence of os penis is sometimes an important indicator of gender identification
[399} Other methods of estimating height at withers may also be used:
A. Chrószcz, M. Janeczek, V. Onar, P. Staniorowski, N. PospiesznyThe shoulder heigh estimation in dogs based on internal dimension of cranial cavity using mathematical formula. Anat. Histol. Embryol. 2007, 36 (4), 269-271 DOI: 10.1111/j.1439-0264.2007.00760.x
A. Chrószcz, M. Janeczek, E. Pasicka, J. Klećkowska-Nawrot. Height at the withers estimation in the horses based on the internal dimension of cranial cavity. Folia Morph. 2014, 73 (2),
. Bestimmung der Höhe im Widerrist bei Pferden. Jahrschrift für Mitteldeutsche Vorgeschichte, 1955, 39, 240– 244.
It would be appropriate to refer to this.
Author Response
We would like to thank the Academic Editor and the Reviewers for the relevant suggestions that have enabled us to improve our manuscript, entitled “Post-mortem Dental Profile as a Powerful Tool in Animal Forensic Investigations—A Review”.
We have carried out the full revisions requested, and we hope that you and the Reviewers will now find our manuscript of sufficient quality and reader interest for publication in Animals. Please find below the full listing of Reviewer comments and indications of the revisions/ modifications that are now incorporated into our revised manuscript. The changes are highlighted in the manuscript using Microsoft Word’s ‘track changes’ editing feature.
COMMENTS TO REVIEWER 3
The work is interesting and addresses a relatively rarely addressed issue.
[80] Injuries may be due to cultural conditions determining the manner of killing.
Reply: This comment was added in the manuscript (Line 112).
[255] Please add that the presence of os penis is sometimes an important indicator of gender identification.
Reply: The reference to the baculum/baubellum for sex estimation was added to the manuscript (Line 323).
[399} Other methods of estimating height at withers may also be used:
Chrószcz, M. Janeczek, V. Onar, P. Staniorowski, N. Pospieszny. The shoulder heigh estimation in dogs based on internal dimension of cranial cavity using mathematical formula. Anat. Histol. Embryol. 2007, 36 (4), 269-271 DOI: 10.1111/j.1439-0264.2007.00760.x
Chrószcz, M. Janeczek, E. Pasicka, J. Klećkowska-Nawrot. Height at the withers estimation in the horses based on the internal dimension of cranial cavity. Folia Morph. 2014, 73 (2),
Müller, H. H. Bestimmung der Höhe im Widerrist bei Pferden. Jahrschrift für Mitteldeutsche Vorgeschichte, 1955, 39, 240– 244.
It would be appropriate to refer to this.
Reply: Both references of Chrószcz et al. were cited in the manuscript. Müller’s text could not be obtained, so it was not cited in the text (Lines 474–476).

Round 2
Reviewer 2 Report
All comments have been addressed as directed
Reviewer 3 Report
The paper can be published